# How Are Infants Suspected to Have Cow’s Milk Allergy Managed? A Real World Study Report

**DOI:** 10.3390/nu13093027

**Published:** 2021-08-30

**Authors:** Yvan Vandenplas, Simona Belohlavkova, Axel Enninger, Pavel Frühauf, Niten Makwana, Anette Järvi

**Affiliations:** 1Vrije Universiteit Brussel (VUB), UZ Brussel, KidZ Health Castle, 1090 Brussels, Belgium; 2Immuno-flow, s.r.o., 12843 Prague, Czech Republic; Simona.Belohlavkova@seznam.cz; 3Olgahospital, Zentrum für Kinder-, Jugend- und Frauenmedizin, Klinikum Stuttgart, 70174 Stuttgart, Germany; A.Enninger@klinikum-stuttgart.de; 4Pediatric Clinics and Inherited Metabolic Disorders, 1st Faculty of Medicine Charles University, 12108 Prague, Czech Republic; fruhaufp@volny.cz; 5Sandwell and West Birmingham Hospitals, Birmingham B18 7QH, UK; nmakwana@nhs.net; 6Nestlé Health Science, 1800 Vevey, Switzerland; anette.jarvi@nestle.com

**Keywords:** cow’s milk allergy, CoMiSS, extensive hydrolysate, partial hydrolysate, amino acid formula

## Abstract

The purpose of this study was to evaluate the diagnosis and management of infants presenting with symptoms attributable to cow’s milk allergy (CMA) in a real life setting and to test how the Cow’s Milk-related Symptom Score (CoMiSS^®^) can be used to support the awareness to diagnose cow’s milk protein allergy in primary care practice. The CoMiSS is an awareness tool based on various symptoms such as crying, gastrointestinal symptoms, dermatological and respiratory symptoms. The study was conducted on 268 infants from four countries (Belgium, Czech Republic, Germany, UK) aged 0 to 18 months consulting for CMA related symptoms. The analysis was based on two visits of these subjects. The results show an average CoMiSS of 11 at the first visit. After a therapeutic dietary intervention, the score at the second visit, which happened 3 weeks ± 5 days after the first one, dropped to an average value of 4. A satisfaction questionnaire completed by the primary care practitioners suggested an overall high level of satisfaction with the application of the CoMiSS tool in routine practice. These data highlight a huge discrepancy in the diagnosis and management of infants suspected of CMA in the different countries. The findings suggest that the CoMISS questionnaire is an effective tool in aiding awareness of CMPA in primary health care.

## 1. Introduction

The risk of developing an allergy has become a significant public health issue with increasing prevalence [1]. The diagnosis of cow’s milk allergy (CMA) can be challenging, since symptoms can be immediate (IgE mediated) as well as delayed (non-IgE mediated), and involve many organ systems. Gastro-intestinal (GI) symptoms attributed to non-IgE mediated CMA include, amongst others, infantile colic, food protein induced enterocolitis syndrome, food protein induced allergic proctocolitis, food allergic enteropathy, eosinophilic disorders, and food protein induced dysmotility disorders, food protein induced constipation, and food protein induced gastro-esophageal reflux (GER) [2]. Cutaneous manifestations, such as deterioration of atopic dermatitis and urticaria, respiratory symptoms and general manifestations such as failure to thrive, distress and crying are part of the spectrum of CMA [3]. CMA is the most common food allergy in childhood and its prevalence ranges from 1.9% to 4.9% [4]. The diagnosis of CMA is suspected after a thorough history and physical examination, including the evaluation of growth [5].

Up to 25 to over 50% of infants develop functional gastro-intestinal disorders (FGIDs) [6,7]. As the spectrum of manifestations of FGIDs and mainly non-IgE mediated CMA do overlap, they may be difficult to separate leading to difficulty in distinguishing from FGIDs, GERD and CMA [8]. As a consequence, the prevalence of CMA is debated. Laboratory tests assist the diagnosis of IgE mediated CMA, but can be negative [5]. A confirmed diagnosis of non-IgE mediated CMA requires an oral food challenge (OFC) after a diagnostic elimination diet of two to four weeks [5]. An open food challenge after the elimination diet is considered an adequate diagnostic tool in clinical practice [5]. The double-blind-placebo-controlled-food challenge, considered as the “gold standard”, is needed to confirm the diagnosis in clinical research [9]. A well performed challenge includes the progressive at-home reintroduction of milk, which can be safely done, especially in children with non-IgE mediated CMA with delayed reactions [9]. The oral food challenge in patients with IgE mediated reactions (Skin Prick Test (SPT) and/or sIgE positive) should be performed under medical supervision. An early diagnosis of CMA is important as a delayed diagnosis may lead to nutritional disorders and as a consequence an increased risk of impaired growth [10]. Moreover, a delayed diagnosis and incorrect management also increases parent and caregiver anxiety and economic cost as the symptoms place a burden on both the infant and their caregivers [11,12].

Infants with allergic disorders are presented to different healthcare professionals (HCPs) spanning multiple specialties (e.g., general practitioners, general pediatricians but also pediatric subspecialists in gastroenterology, allergy/immunology, dermatology) with diverse levels of expertise [1]. As a consequence, there is a great variability in dietary management approaches [1]. The purpose of this study was to assess in a real life situation the diagnossis and management of infants presenting with symptoms attributable to CMA, and to test the usefulness of the Cow’s Milk-related Symptom Score (CoMiSS^®^) (Table 1).

## 2. Materials and Methods

A multicentre prospective observational single cohort study was carried out by 84 HCPs between September 2016 and September 2018. Recruiting sites were located in four European countries including Belgium, Germany, Czech Republic and the United Kingdom (UK). The study was approved by the Ethics Committee and regulatory authority, where applicable. Children were enrolled by independent primary health care practitioners in Belgium and Germany. In the Czech Republic, inclusion was done by primary health care and allergologists. Due to the specific structure of the medical organization in the UK, recruitment was conducted in four specialized pediatric care centers.

Infants of both sexes, of any ethnicity, aged 0 to 18 months suspected of mild to moderate symptoms of CMA as primary clinical impression of the practitioner were consecutively enrolled. Subjects having congenital disease or malformations, significant pre-natal or post-natal diseases, subjects with minor parents or parents who could not comply with study procedures and subjects included in other clinical trials were excluded. Prior to enrolment, a written informed consent was obtained from both parents, or one parent in single-parent families. The study design included two visits: at baseline and after 3 weeks ± 5 days (Figure 1: study flow chart).

The CoMiSS score was determined by the HCP during both visits using the CoMiSS awareness tool form. At baseline, basic information such as date of birth, sex, weight, length and head circumference were recorded, as well as the history of CMA-related symptoms, such as duration of symptoms and diet at baseline. The dietary and medical intervention recommended by the HCP was registered. Information was also recorded regarding the requests of HCPs for investigations such as blood sampling and skin prick testing for diagnostic purposes.

At the end of the study each practitioner was asked to complete a satisfaction questionnaire about the use of CoMiSS awareness tool.

## 3. Results

Two hundred and sixty-eight subjects (145 boys/117 girls/6 unknown) were found eligible and enrolled and are reported as the intent-to-treat (ITT) population (Table 2. Baseline characteristics; Table 3: Age at inclusion).

Out of 268 enrolled subjects, 84 were recruited from Czech Republic, 84 from Belgium, 36 from Germany and 54 from the UK. The final visit was between 16 and 26 days after baseline visit in 208 infants and is reported as the per protocol (PP) population. Among the 268 infants, 16 did not make the final visit, and in 44 the final visit was either less than 16 days or more than 26 days post-baseline visit. Preliminary analysis showed no clinically meaningful difference in the results between the ITT and the PP population; therefore data according to the ITT population analysis are reported

The mean duration of symptoms (Table 4) was 12 weeks and ranged between 7 weeks in Belgium to 24 weeks in the UK.

At baseline visit, 31% (83/268) of the included infants were exclusively breastfed, 33% (88/268) were formula-fed, 31% (and 30% (80/268) were on mixed breastfeeding and formula feeding (Data missing of 17/268 (6%) infants). Further, 56% of the formula fed infants were fed standard infant formula, 21% partially hydrolysed formula (pHF), 11% an extensively hydrolysed formula (eHF) and 7% were fed amino acid formula (AAF) formula (Table 5: Feeding per country).

The time between the introduction of cow’s milk formula and the onset of symptoms was recorded in only 37% (100/268) of subjects. According to the data, the time interval between ingestion of cow’s milk and the onset of symptoms in the UK was only a few hours compared to a broad range of 0 up to 90 days in the other countries (Table 6: Time interval between ingestion of cow’s milk and appearance of symptoms.

The CoMiSS was collected from the 268 subjects at the baseline visit. The score ranged from 0 to 28 (Figure 2).

Overall, the mean and median CoMiSS was 11.1 and 11.0, respectively, and 72.3% of subjects had a CoMiSS of >9 and 48.9% a CoMiSS ≥ 12. The CoMiSS score was lowest in the UK. Stratification of CoMiSS according to the cut-off value of 12 divided the infants in two roughly equal groups. However, country-specific CoMiSS stratification reflected that the majority of British and German subjects had scores below <12 (78% and 81%, respectively), while the majority of the Czech infants (82%) had CoMiSS of ≥12 (Table 7).

A cow’s milk elimination diet was prescribed in 36% of the breastfeeding mothers (exclusive and mixed breastfeeding, n:164) and in 59% of the formula fed infants. An elimination diet in a breastfeeding mother was almost twice as frequently recommended if the CoMiSS was ≥12 than if CoMiSS was < 12 (24.8 vs. 47.3%, respectively). An eHF and an AAF were prescribed almost equally to 31% of formula fed infants (Table 8). An AAF was prescribed almost twice as frequently in subjects having a CoMiSS ≥ 12 than subjects having CoMiSS <12 (19.7% vs. 42.7%, respectively). Prescription of an eHF appeared almost equal for subjects that had CoMiSS < 12 and ≥12 (~30%). A pHF was recommended in ~6%, including 8% in the group with a CoMiSS ≥ 12.

The prescription rate of pHF differed per country: 0% in the UK, 3% in Germany, 5% in Belgium and 11.9% in the Czech Republic. The prescription rate of eHF and AAF also differed from country to country (Table 9). In the Czech Republic, eHF was prescribed to only 8.3% of the infants, and AAF was recommended in 72.6%. In Germany, eHF and AAF prescription was comparable (27.8 vs. 22.2%, respectively). In Belgium and the UK, an eHF was much more frequently recommended than an AAF (54.8 and 25.5% vs. 8.6 and 10.9%, respectively.

The CoMiSS at the final visit was obtained in 246/268 (94%) subjects, and was significantly lower suggesting efficacy of the therapeutic actions taken (Figure 3: CoMiSS after intervention at final visit).

The overall mean of CoMiSS decreased from 11.1 at baseline to 4.2 at the final visit; the median was reduced from 11 to 4.0. At the final visit, only four infants had a CoMiSS ≥12, and 23 infants had a score of > 9. In formula fed infants, the change in CoMiSS was greater for AAF fed infant than for eHF fed infants (Table 10). The dietary intervention resulted in a significant decrease of the CoMiSS in the vast majority of infants. There was almost no difference between the change of the mean or median CoMiSS. In exclusively breastfed infants, the median CoMiSS decreased by 6.0 during the elimination diet, while the decrease was 11.0 in partially breastfed infants to whom also eHF or AAF was prescribed. Finally, the decreased score in formula fed infants observed with eHF was lower than with AAF (−6.0 vs. −10.0, respectively). A further analysis excluding infants who were on an eHF or AAF at inclusion did not result in a different outcome, as the number of infants on these formulas at inclusion was very low.

An open oral food challenge to confirm the diagnosis of CMA was performed in 17 infants, and was positive in 4 (24%). At the baseline visit, blood sampling, including IgE tests were much more frequently requested in infants with a CoMiSS ≥12. On the contrary, skin prick tests were less frequently requested in the group with CoMiSS ≥12. There is a large difference in performed diagnostic investigations according to country (Table 11). In Germany skin prick tests were not performed, whilst up to 17% of the infants in the Czech Republic had a skin prick test for cow′s milk. Specific IgE for cow′s milk was measured in almost half the children in the UK and Czech Republic (49.1 and 50.7%, respectively) whilst only 8.3% had this performed in Germany.

At the end of the study, 77/84 (91.6%) health care providers completed the satisfaction questionnaire. Approximately 3 in 5 agreed that the CoMiSS was helpful to consider the diagnosis of CMA more rapidly. Seventy percent intended to continue using the CoMiSS tool and 77% would recommend CoMiSS to their colleagues. About 25% mentioned that the CoMiSS tool lengthens consultation time (Table 12).

## 4. Discussion

The CoMiSS was initially developed as the Symptom Based Score and was intended to facilitate comparability of the efficacy of two extensively hydrolyzed formulas in patients suspected of CMA, and included in a prospective, randomized, double-blind trial [13]. A group of key opinion leaders suggested this tool could be used as an awareness tool in order to increase the awareness of the most common symptoms of CMA to aid an earlier diagnosis [3]. This study confirms that the CoMiSS can be considered as a useful awareness tool for HCPs, what was already previously suggested in a review paper [14].

This real world collection of data highlights the differences in baseline characteristics in infants suspected to suffer from CMA per country, which can be related in part to the varied health care system practices (Table 3, Table 4, Table 5, Table 6 and Table 7). Therefore, the second visit was scheduled after 3 weeks, considering the recommendations for a diagnostic elimination diet during 2 to 4 weeks and local health care organization habits. This was an observational study, whose primary objective was to describe the diagnostic and therapeutic actions taken both overall and stratified by baseline CoMiSS in a general pediatric population consulting for symptoms possibly related to CMPA. The median age at inclusion was 18.4 weeks, but differed from 8.7 weeks in Belgium to 32.4 weeks in the UK, and, as a consequence, a large discrepancy in the mean duration of symptoms was observed, varying from 4.0 weeks in Belgium to 21.4 weeks in the UK. Moreover, a large difference in time between ingestion of cow′s milk and appearance of symptoms was reported according to the country, and ranged from 0.5 h in the UK, over 60 h in Belgium and Czech Republic, to as long as 168 h in Germany. Systemic blood sampling was not part of this observational study (as it was the goal to highlight differences according to country), but the differences in baseline characteristics suggest that the children included in the UK presented mostly with IgE mediated allergy because of the short lapse of time between ingestion of cow’s milk and appearance of symptoms, while the German infants mainly have non-IgE mediated allergy. These huge discrepancies in baseline characteristics contribute to a better understanding of the discrepancies in diagnosis, management and outcome of CMA according to country, and thus health care system. These baseline differences in population are likely to explain the difference in CoMiSS scoring between countries. About 50% of the infants had a baseline CoMiSS ≥ 12, but this ranged from 19.4% in Germany to 82.1% in the Czech Republic (Table 8). Initially, an arbitrarily decided cut-off value of ≥12 had been proposed to predict the likelihood of CMA (3). A score of 12 requires the presence of a minimum of two severe symptoms and a score of >12 requires the presence of at least three symptoms and two organ systems [3]. Subsequent evidence from literature, both in a supposed healthy population and in symptomatic infants suggest that a cut-off of > 9 might be more appropriate [15,16,17]. Therefore, a further study exploring the efficiency of cut-off > 9 might be of interest.

A pHF was recommended as a therapeutic intervention in 6% of the infants, which is in disagreement with all existing guidelines [1]. The difference in prescription rate of pHF per country, ranging from 0% in the UK to 11.9% in the Czech Republic also illustrates the differences in education and training of the HCPs across Europe. Variation was also observed in prescription rates of eHF and AAF according to country. Differences in population selection, education and training as well as differences in reimbursement systems may contribute to these discrepancies. In the Czech Republic, there is full reimbursement of eHF and AAF, if prescribed by allergologists and pediatric gastroenterologists; general practitioners can only obtain reimbursement for 5 packages of AAF per patient. In Germany, there is full reimbursement in case of demonstrated IgE mediated allergy, while in non-IgE-mediated allergy, resolution of symptoms after 2 weeks of cow’s milk exclusion needs to demonstrated and full reimbursement is obtained if the efficacy is well documented. In Belgium, only AAF is (almost) fully reimbursed, on condition allergy to an eHF is demonstrated or if symptoms are severe (failure to thrive, anaphylaxis). Inconsistencies in the diagnostic investigations performed per country again illustrate the differences in health care systems practices between countries. Since in the UK much more children are included with immediate type of reactions, suggesting IgE mediated allergy, it is logical that IgE levels are much more frequently determined than in countries such as Germany where the vast majority of children have delayed reactions, indicating non-IgE mediated allergy (Table 10). The discrepancy between UK and Germany in IgE and non IgE CMA can probably also be explained by difference in study sites, since in Germany mainly gastroenterologists did collaborate (and symptoms involving the GI tract are mainly non-IgE) while in the UK allergists were involved, who do see more frequently IgE mediated symptoms.

The elimination diet resulted in a decrease of 9 points or more in 51% of the infants (Table 11). In formula-fed infants, the decrease with AAF was larger than with eHF. According to this observational study, prescription of an AAF formula was the only factor with a relevant effect. As this is an open observational study, interpretation on efficacy may be biased. ESPGHAN and other guidelines recommend that the first line of prescription should be an eHF. However, this study suggested that primary HCPs frequently recommended AAF as the first formula. The reasons why this is observed has to be further investigated, as this could be due to differences in education and training, patient selection, availability of products, and reimbursement, which may all influence the choices made by the HCP.

## 5. Conclusions

This observational study suggests that CoMiSS is an efficient and reliable tool to facilitate awareness of the diagnosis of CMA in real life for HCPs. These results also show the discrepancies between countries, as baseline characteristics differ substantially according to country. It is very likely that the practice between different health care systems contributes to the variation observed in patient selection, diagnostic procedures and management.

## Figures and Tables

**Figure 1 nutrients-13-03027-f001:**
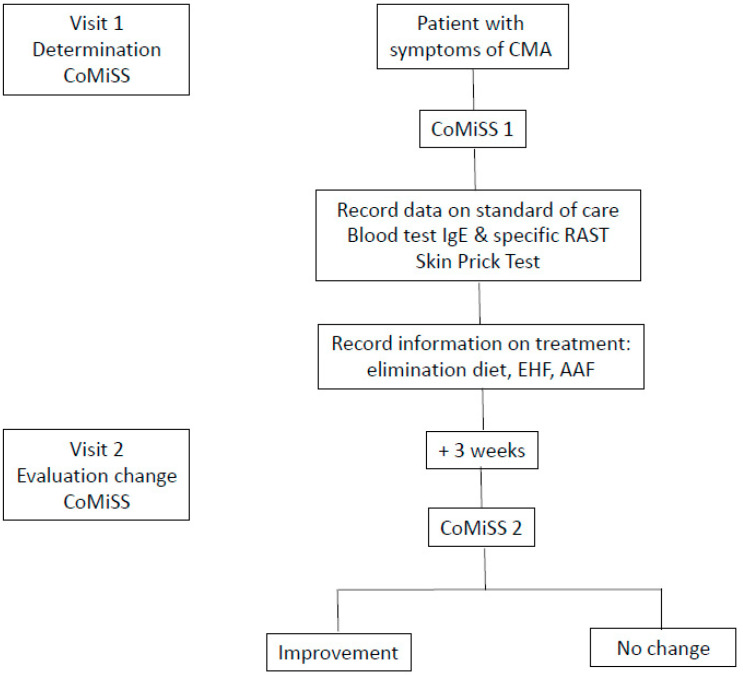
Study flow chart.

**Figure 2 nutrients-13-03027-f002:**
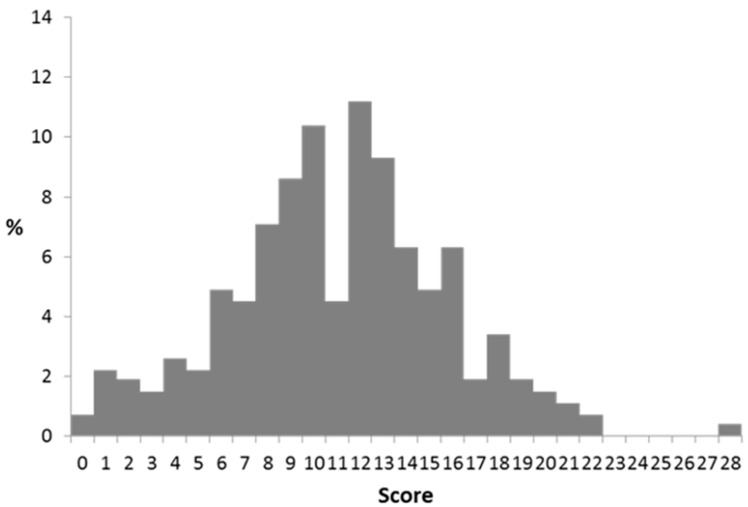
CoMiSS distribution at baseline.

**Figure 3 nutrients-13-03027-f003:**
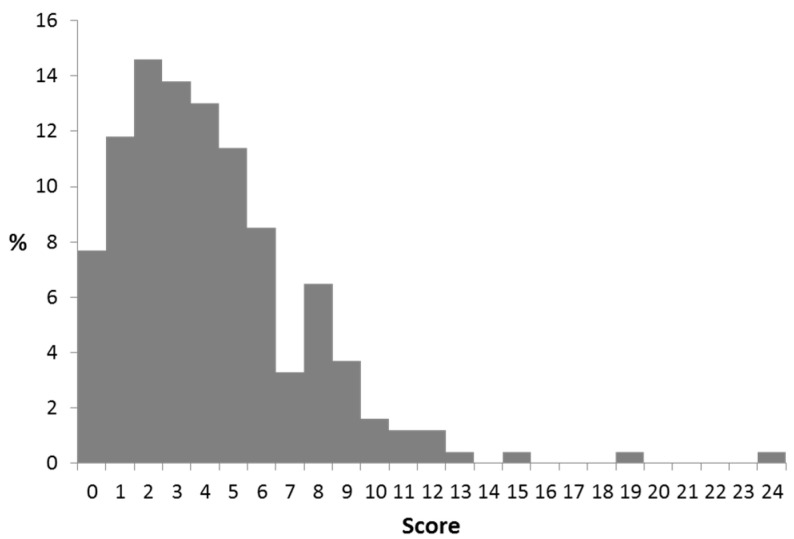
CoMiSS after intervention at final visit.

**Table 1 nutrients-13-03027-t001:** The Cow’s Milk-related Symptom Score (CoMiSS^®^) [3].

Symptom	Score			
Crying	0	≤1 h/day		
	1	1 to 1.5 h/day		
	2	1.5 to 2 h/day		
	3	2 to 3 h/day		
	4	3 to 4 h/day		
	5	4 to 5 h/day		
	6	≥5 h/day		
Regurgitation	0	0 to 2 episodes/day		
	1	≥3 to ≤5 episodes of small volume		
	2	>5 episodes of >1 coffee spoon		
	3	>5 episodes of ±half of the feedings in <half of the feedings		
	4	continuous regurgitations of small volumes >30 min after each feeding		
	5	regurgitation of half to complete volume of a feeding in at least half of the feedings		
	6	regurgitation of the complete volume after each feeding		
Stools (Bristol scale)	4	type 1 and 2 (hard stools)		
	0	type 3 and 4 (normal stools)		
	2	type 5 (soft stool)		
	4	type 6 (liquid stool, if unrelated to infection)		
	6	type 7 (watery stools)		
Skin symptoms	0 to 6	Atopic eczema	Head-neck-trunk	Arms-legs-hands-feet
		Absent	0	0
		Mild	1	1
		Moderate	2	2
		Severe	3	3
	0 to 6	Urticaria (0: no, 6: yes)		
Respiratory symptoms	0 1 2 3	no respiratory symptoms slight symptoms mild symptoms severe symptoms		

**Table 2 nutrients-13-03027-t002:** Baseline characteristics of the 268 included infants.

	Mean	SD	Median	Min	Max	N
Age (weeks)	21.9	16.3	18.4	1.4	80.6	265
Weight (kg)	6.4	2.0	6.3	3.1	13.1	251
Length (cm)	62.7	8.3	62.5	37.0	88.6	231
Duration of symptoms(weeks)	12.1	12.9	7.4	0.0	64.1	255

Legend: SD: standard deviation; Min: minimum; Max: Maximum; n: number of patients for whom information was available.

**Table 3 nutrients-13-03027-t003:** Age distribution of included infants per country.

	Age (Weeks)					
	Mean	SD	Median	Min	Max	N
Czech Republic	24.1	14.2	22.6	1.4	65.1	84
Germany	21.4	17.7	16.4	3.7	79.4	36
Belgium	12.7	10.0	8.7	2.4	53.4	90
UK	34.1	17.7	32.1	4.0	80.6	55
All	21.9	16.3	18.4	1.4	80.6	265 °

Legend: SD: standard deviation; Min: minimum; Max: Maximum; °: age of 3 infants missing.

**Table 4 nutrients-13-03027-t004:** Duration of symptoms before inclusion per country.

	Duration of Symptoms (Weeks)					
	Mean	SD	Median	Min	Max	N
Czech Republic	11.5	10.4	10.5	0.0	49.0	80
Germany	9.7	12.4	5.9	0.0	49.6	35
Belgium	6.9	9.1	4.0	0.0	56.1	90
UK	24.0	15.4	21.2	1.6	64.1	50
All	12.1	12.9	7.4	0.0	64.1	255 °

Legend: SD: standard deviation; Min: minimum; Max: Maximum; °: data missing for 13 infants.

**Table 5 nutrients-13-03027-t005:** Type of formula at inclusion visit in formula-fed or mixed breastfeeding and formula-fed infants.

	Type of Formula									
	SIF		pHF		eHF		AAF		Other	
	n	%	N	%	n	%	n	%	n	%
Czech R	25	47.2	19	35.8	2	3.8	5	9.4	2	3.8
Germany	13	50.0	9	34.6	2	7.7	2	7.7	0	0.0
Belgium	52	72.2	9	12.5	5	6.9	1	1.4	5	6.9
UK	13	38.2	2	5.9	11	32.4	4	11.8	4	11.8
All °	103	55.7	39	21.1	20	10.8	12	6.5	11	5.9

Legend: %: percent; n: number; SIF: standard infant formula; pHF; partially hydrolyzed formula; eHF: extensively hydrolyzed formula; AAF: amino acid formula; R: Republic; °: data for 185 infants (+83 partially breastfed infants) = 268 infants).

**Table 6 nutrients-13-03027-t006:** Time between first intake of cow’s milk and onset of symptoms per country.

	Time between Ingestion of Cow’s Milk and Onset of Symptoms (Hours)					
	Mean	SD	Median	Min	Max	N
Czech Republic	219.8	431.4	60.0	0.1	2160.0	28
Germany	321.6	365.6	168.0	0.0	1008.0	10
Belgium	247.6	339.6	60.0	0.0	1080.0	38
UK	3.4	7.3	0.5	0.0	24.0	24
All	188.6	343.1	24.0	0.0	2160.0	100

Legend: SD: standard deviation; Min: minimum; Max: Maximum; n: number.

**Table 7 nutrients-13-03027-t007:** CoMiSS < and >12 distribution per country at baseline and final visit.

	CoMiSS			
	At Baseline		Final Visit	
	<12	≥12	<12	≥12
	n (%)	n (%)	n (%)	n (%)
Czech R	15 (17.9)	69 (82.1)	82 (98.8)	1 (1.2)
Germany	29 (80.6)	7 (19.4)	32 (97.0)	1 (3.0)
Belgium	50 (53.8)	43 (46.2)	87 (97.8)	2 (2.2)
UK	43 (78.2)	12 (21.8)	38 (92.7)	3 (7.3)
All	137 (51.1)	131 (48.9)	239 (97.2)	7 (2.8)

Legend: n: number; %: percent; R: republic.

**Table 8 nutrients-13-03027-t008:** Actions undertaken at first visit, stratified by CoMiSS at baseline.

Action	CoMiSS Score at First Visit				All	
	<12		≥12			
	n	%	n	%	N	%
Elimination diet mother °	34	24.8	62	47.3	96	35.8
Elimination diet child *	68	49.6	91	69.5	159	59.3
pHF prescribed	5	3.6	11	8.4	16	6.0
eHF prescribed	42	30.7	40	30.5	82	30.6
AAF prescribed	27	19.7	56	42.7	83	31.0

Legend: pHF: partially hydrolysed formula; eHF: extensively hydrolysed formula; AAF: amino acid formula. °: exclusive and partial breastfeeding combined; *: partial breastfeeding and full formula feeding.

**Table 9 nutrients-13-03027-t009:** Actions undertaken at first visit, stratified by CoMiSS at baseline, per country.

	CoMiSS at First Visit				All	
	<12		≥12			
	n	%	n	%	N	%
Czech Republic						
Elimination diet mother	7	46.7	34	49.3	41	48.8
Elimination diet child	12	80.0	53	76.8	65	77.4
pH formula	1	6.7	9	13.0	10	11.9
eH formula	0	0.0	7	10.1	7	8.3
AA formula	12	80.0	49	71.0	61	72.6
Germany						
Elimination diet mother	5	17.2	2	28.6	7	19.4
Elimination diet child	12	41.4	6	85.7	18	50.0
pH formula	1	3.4	0	0.0	1	2.8
eH formula	6	20.7	4	57.1	10	27.8
AA formula	6	20.7	2	28.6	8	22.2
Belgium						
Elimination diet mother	11	22.0	18	41.9	29	31.2
Elimination diet child	27	54.0	29	67.4	56	60.2
pH formula	3	6.0	2	4.7	5	5.4
eH formula	25	50.0	26	60.5	51	54.8
AA formula	3	6.0	5	11.6	8	8.6
UK						
Elimination diet mother	11	25.6	8	66.7	19	34.5
Elimination diet child	17	39.5	3	25.0	20	36.4
pH formula	0	0.0	0	0.0	0	0.0
eH formula	11	25.6	3	25.0	14	25.5
AA formula	6	14.0	0	0.0	6	10.9

Legend: n: number; %: percent; pH: partial hydrolysate; eH: extensive hydrolysate; AA: amino acid.

**Table 10 nutrients-13-03027-t010:** Dietary intervention and change in CoMiSS.

Intervention	Change in CoMiSS Score at Final Visit					
	Mean	SD	Median	Min	Max	n
Elimination diet mother	−5.8	5.0	−6.0	−17.0	12.0	96
Partial breastfed and eHF or AAF	−10.0	6.2	−11.0	−17.0	0.0	6
eHF	−6.4	5.1	−6.0	−19.0	5.0	70
AAF	−9.5	4.5	−10.0	−27.0	−1.0	74

Legend: eHF: extensively hydrolysed formula; AAF: amino acid based formula; SD: standard deviation; n: number.

**Table 11 nutrients-13-03027-t011:** Diagnostic actions requested stratified by country (%).

	SPT	sIgE
Czech Republic	17.0	50.7
Germany	0	8.3
Belgium	15.1	31.2
UK	3.6	49.1

Legend: SPT: skin prick test for cow’s milk; sIgE: specific IgE for cow’s milk;%: percent of infants.

**Table 12 nutrients-13-03027-t012:** Satisfaction questionnaire for health care provider.

	Questions	Response Health Care Provider (%)				
		Fully Agree	Agree	=	Disagree	Strong Disagree
1	The time it took you on average to complete the CoMiSS tool did NOT significantly lengthen the total consultation time.	18	45	12	20	5
2	Based on the experience of your use of the CoMiSS tool, you think/believe that this tool is helpful in the diagnosis of CMPA.	25	61	9	5	0
3	You think/believe that the tool can help physicians to diagnose infants with CMPA faster.	17	64	13	6	0
4	You intend to continue using the CoMiSS tool in your practice.	21	49	18	9	3
5	You would recommend the tool to your colleagues.	19	57	17	4	3

Legend: =: neither agree or disagree.

## Data Availability

No additional data.

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
