# Peer review of "How Are Infants Suspected to Have Cow’s Milk Allergy Managed? A Real World Study Report"

_nutrients, 2021, doi:10.3390/nu13093027_

Round 1
Reviewer 1 Report
The Authors of the article present recommendations for the introduction of a questionnaire Cow Milk-related Symptom Score (CoMiSS) assessing the effectiveness of infant qualifications and introduced dietary therapies in allergy to cow's milk. The conducted research performed in four European countries (Belgium, Germany, the Czech Republic and Great Britain) concerns a group of 268 newborns showing food intolerance to cow's milk. The results of their research show the diversity of the nature of intolerance depending on the country health care system, treatment of the infant and the type of diet used. Data analysis shows that the use of the survey (CoMiSS) can support the awareness to diagnose cow’s milk protein allergy in primary care practice.
Minor revision:
Please correct few errors:
line 60 – word repetition (under)
line 248 – capital letter (Republic)
line 278 – no spaces (of the)
line 289 – contain instead containe
Author Response
Dear reviewer
We thank the reviewer for the positive comments and suggestions.
|
line 60 – word repetition (under) |
corrected |
|
line 248 – capital letter (Republic) |
Done |
|
line 278 – no spaces (of the) |
Done |
|
line 289 – contain instead containe |
Done |
Reviewer 2 Report
The purpose of this observational study was to assess the efficiency of the Cow’s Milk-related Symptom Score (CoMiSS) in the diagnostic and management of infants with cow milk allergy (CMA) related symptoms. Huge discrepancies in in CMA diagnostic and management criteria were observed among the four European countries involved in the study. Using a convenience cutoff >12 to predict the likelihood of CMA, the CoMiSS was shown to be efficient in diagnosis and management. A positive response in using the CoMiSS was obtained in a questionnaire of satisfaction to involved health care professionals (HCPs).
The study is interesting, but some aspects need clarification.
Major queries
- Abstract (lines 15-18): the purpose statement is confusing, including the meaningless repetition of “diagnosis and management”. Please rephrase.
- Lines 69-72: the purpose statement is confusing, including the meaningless repetition of “approach”. Please rephrase.
- Lines 83-84: pleased specify the method that was used for participants’ selection and enrolment (random? consecutive cases? by convenience?)
- Lines 89: pleased specify the rationale for choosing a mean of 3 weeks after baseline (and not other time) for the second visit.
- Lines 133-124: the attrition rate due to missing records of the time between cow’s milk formula introduction and the onset of symptoms was 63%. It should be discussed if this high percentage of loss to follow-up may have introduced some bias (Kristman 2004).
- Please explain the reason for not using statistical analysis to assess the significance of differences in comparisons shown in Tables 8 to 11, particularly between CoMiSS < and ≥ 12?
- Lines 206-206: the statement is confusing, particularly the meaning of “was helpful to enable considering the diagnosis”. Please rephrase.
- Line 268: other term than “significant” should be used because the effect of prescription of an AAF formula was not assessed through statistical analysis
- Line 278: primary and specialized HCPs were involved in the study. Therefore, conclusions should not be confined to primary HCPs.
- Evidence from literature suggests that a CoMiSS threshold 9 may be more appropriate for diagnosis and management approach of infants with CMA-related symptoms. Therefore, in Discussion the authors should consider that a further study exploring the efficiency of cut-off ≥9 might is needed.
Minor queries
- Line 59: explain the abbreviation “SPT” that appears for the first time
- Correct typo errors: line 178: “visit” instead of “vist”, line 237 “DMA” instead of “CMPA”
Reference
Kristman V, Manno M, Côté P. Loss to follow-up in cohort studies: how much is too much? Eur J Epidemiol. 2004;19(8):751-60. doi: 10.1023/b:ejep.0000036568.02655.f8.
Author Response
Dear Reviewer
We thank you for your positive comments and suggestions regarding our paper.
|
· Abstract (lines 15-18): the purpose statement is confusing, including the meaningless repetition of “diagnosis and management”. Please rephrase. |
rephrased |
|
Lines 69-72: the purpose statement is confusing, including the meaningless repetition of “approach”. Please rephrase. |
rephrased |
|
· Lines 83-84: pleased specify the method that was used for participants’ selection and enrolment (random? consecutive cases? by convenience?) |
Consecutive - changed in the manuscript |
|
· Lines 89: pleased specify the rationale for choosing a mean of 3 weeks after baseline (and not other time) for the second visit. |
We added a sentence in the discussion section: " Therefore, the second visit was scheduled after 3 weeks, considering the recommendations for a diagnostic elimination diet during é to 4 weeks and local health care organization habits." |
|
Lines 133-124: the attrition rate due to missing records of the time between cow’s milk formula introduction and the onset of symptoms was 63%. It should be discussed if this high percentage of loss to follow-up may have introduced some bias (Kristman 2004). |
We would like to clarify the sentence: this is not a loss to or in follow-up. At the moment of inclusion, it was tried to collect a posteriori information on the time that elapsed between first introduction of cow milk and the appearance of symptoms. Only 37% of the parents could provide this information. |
|
Please explain the reason for not using statistical analysis to assess the significance of differences in comparisons shown in Tables 8 to 11, particularly between CoMiSS < and ≥ 12? |
Please find below the answer of the statistician. We added this information in the discussion section.
This was an observational study, whose primary objective was to describe the diagnostic and therapeutic actions taken both overall and stratified by baseline CoMiSS score in a general pediatric population consulting for symptoms possibly related to CMPA. Analyses of the kind suggested were not planned. Therefore, if tests of significance were conducted, they would be post-hoc and thus invalid. Additionally, the actions taken and dietary interventions listed in Tables 9-11 are not mutually exclusive. |
|
· Lines 206-206: the statement is confusing, particularly the meaning of “was helpful to enable considering the diagnosis”. Please rephrase. |
Adapted |
|
· Line 268: other term than “significant” should be used because the effect of prescription of an AAF formula was not assessed through statistical analysis |
We changed "significant" to "relevant". |
|
· Line 278: primary and specialized HCPs were involved in the study. Therefore, conclusions should not be confined to primary HCPs. |
We deleted "primary". |
|
Evidence from literature suggests that a CoMiSS threshold 9 may be more appropriate for diagnosis and management approach of infants with CMA-related symptoms. Therefore, in Discussion the authors should consider that a further study exploring the efficiency of cut-off ≥9 might is needed. |
We added the suggestion of the reviewer in the discussion section. |
|
Line 59: explain the abbreviation “SPT” that appears for the first time |
Done |
|
Correct typo errors: line 178: “visit” instead of “vist”, line 237 “DMA” instead of “CMPA” |
Corrected |
|
Kristman V, Manno M, Côté P. Loss to follow-up in cohort studies: how much is too much? Eur J Epidemiol. 2004;19(8):751-60. doi: 10.1023/b:ejep.0000036568.02655.f8. |
|
Round 2
Reviewer 2 Report
The revised manuscript is improved, but there are still minor aspects and typo errors that should be addressed or corrected:
- As stated in the Author’s reply, statistical analysis was not performed because the objective of this observational study was to describe the diagnostic and therapeutic actions taken both overall and stratified by baseline CoMiSS in a pediatric population with symptoms possibly related to CMPA. Accordingly, the objective should be more accurately stated, replacing the term “test” with “describe” (lines 16 and 70), since testing presupposes an analytical study supported with statistical analysis, and not a pure descriptive study.
- The Authors explained that the attrition rate was not addressed because “information on the time elapsed between first introduction of cow milk and appearance of symptoms was collected a posteriori at the moment of inclusion”. This is an important detail that should be included in Materials and Methods section.
- The term “score” is redundant wherever it appears after stating COMISS, since the last “S” of this abbreviation means score (lines 91, 143, 145, and first line of Tables 9 and 11)
- Line 69: “diagnosis” should be replaced with “diagnosis”
- Line 222: please correct the typo error “é to 4 weeks”
- Line 296: “Yes” is redundant and should be eliminated